# Radioimmunotherapy of methicillin-resistant *Staphylococcus aureus* in planktonic state and biofilms

B. van Dijk[1], K. J. H. Allen[2], M. Helal[2], H. C. Vogely[1], M. G. E. H. Lam[3], J. M. H. de Klerk[4], H. Weinans[1,5], B. C. H. van der Wal[1], E. Dadachova[2]*

1 Department of Orthopedics, University Medical Center Utrecht, Utrecht, The Netherlands, 2 College of Pharmacy and Nutrition, University of Saskatchewan, Saskatoon, Canada, 3 Department of Radiology and Nuclear Medicine, University Medical Center Utrecht, Utrecht, The Netherlands, 4 Department of Nuclear Medicine, Meander Medical Center Amersfoort, Amersfoort, The Netherlands, 5 Department of Biomechanical engineering, TU Delft, Delft, The Netherlands

* ekaterina.dadachova@usask.ca

**Data Availability Statement:** The data underlying the results presented in the study is available in supporting information file.

## Abstract

### Background

Implant associated infections such as periprosthetic joint infections are difficult to treat as the bacteria form a biofilm on the prosthetic material. This biofilm complicates surgical and antibiotic treatment. With rising antibiotic resistance, alternative treatment options are needed to treat these infections in the future. The aim of this article is to provide proof-of-principle data required for further development of radioimmunotherapy for non-invasive treatment of implant associated infections.

### Methods

Planktonic cells and biofilms of Methicillin-resistant *staphylococcus aureus* are grown and treated with radioimmunotherapy. The monoclonal antibodies used, target wall teichoic acids that are cell and biofilm specific. Three different radionuclides in different doses were used. Viability and metabolic activity of the bacterial cells and biofilms were measured by CFU dilution and XTT reduction.

### Results

Alpha-RIT with Bismuth-213 showed significant and dose dependent killing in both planktonic *MRSA* and biofilm. When planktonic bacteria were treated with 370 kBq of $^{213}$Bi-RIT 99% of the bacteria were killed. Complete killing of the bacteria in the biofilm was seen at 185 kBq. Beta-RIT with Lutetium-177 and Actinium-225 showed little to no significant killing.

### Conclusion

Our results demonstrate the ability of specific antibodies loaded with an alpha-emitter Bismuth-213 to selectively kill *staphylococcus aureus* cells in vitro in both planktonic and biofilm

**Funding:** This research was funded by Health Holland, which is non for profit organization, financed by the Netherlands Organization for Scientific Research (NWO), Grant number LSHM-17026. The funders had no role in study design, data collection and analysis, decision to publish, or preparation of the manuscript.

**Competing interests:** The authors have declared that no competing interests exist.

state. RIT could therefore be a potentially alternative treatment modality against planktonic and biofilm-related microbial infections.

## Introduction

Total joint replacement is the last resort treatment for degenerative joint disease. A feared complication is prosthetic joint infection (PJI) with an incidence of 1–2% after primary hip arthroplasty and 1–4% after primary knee arthroplasty [1]. PJI is difficult to treat as the bacteria form a biofilm on the prosthetic material. This hinders the host immune system, but more important, the bacteria in a biofilm are mostly in a dormant state and therefore not susceptible to most antibiotics [2]. Alpha or beta radiation could potentially damage or destroy these dormant cells because, in contrary to antibiotics, the damaging effects are independent of the cell's metabolic state. However, due to the limited tissue penetration of both alpha and beta radiation it is crucial to get the radionuclide in close vicinity to the cells. Radioimmunotherapy (RIT) relies on the antigen-binding characteristics of the monoclonal antibodies (mAbs) to deliver cytotoxic radiation to target cells and is successfully used in oncology [3]. As microbes express antigens that are unique and different from host antigens, they can be targeted with high specificity and low cross-reactivity. In the past we demonstrated that fungal cells could be eliminated in vitro and in vivo with the radiolabeled microorganism-specific mAbs [4], and later expanded this approach to other fungal and bacterial pathogens such as *Streptococcus pneumoniae* and *Bacillus anthracis* as well as HIV [reviewed in 5]. This implies that bacterial infections of the prosthetic joints can also, in principle, be treated with RIT. The hypothesis underlying the current study is that radioisotopes Lutetium-177 ($^{177}$Lu; a beta-emitter), and Actinium-225 ($^{225}$Ac; an alpha-emitter) or Bismuth-213 ($^{213}$Bi; an alpha-emitter) are able to eradicate *Staphylococcus aureus* using RIT with mAbs directed towards the bacterial cell wall and the biofilm. *S. aureus* is the most common pathogen involved in PJI [6] and therefore this proof-of-principle data is required for further development of RIT for non-invasive treatment of PJIs.

## Materials and methods

### Growth of bacterial cultures

A methicillin-resistant AH4802-LAC strain of *Staphylococcus aureus* [7] was a kind gift from Dr. A.R. Horswill, Professor of Immunology & Microbiology at the University of Colorado, CO, USA. This strain is a known biofilm former on diverse surfaces. For both planktonic growth and biofilm formation, the bacteria were transferred from the frozen stock onto blood agar plates (Tryptic Soy Agar (TSA) with 5% sheep blood) and aerobically cultured overnight at 37˚C. After incubation, 3–4 single colonies were emulsified in tryptic soy broth (TSB) and incubated overnight at 37˚C with agitation (150–200 RPM).

For planktonic growth, the cultures were vortexed for 30 seconds after incubation and thereafter diluted 1:100 in TSB. Bacteria were grown for 3–4 hours until logarithmic phase was reached. The cultures were vortexed for 1 min and measured on a microplate reader (Spectra MAX 250, Molecular Devices, USA) at 600 nm. The cells were washed twice and re-suspended in sterile phosphate buffered saline (PBS). The diluted bacteria were vortexed for 10 seconds after which 100 μl of this suspension was added to the appropriate number of wells of a sterile flat-bottomed 96-well polystyrene tissue culture-treated microtiter plate with a lid (Fisher Scientific).

Biofilm formation was standardized and based on the recommendations described by Stepanović et al. [8]. After initial incubation, the culture was vortexed for 30 seconds and thereafter diluted 1:100 in TSB supplemented with 1% glucose to reach approximately $10^6$ colony forming units (CFU)/ml, measured at 600 nm. The diluted bacteria were vortexed for 10 seconds after which 100 μl of this suspension was added to the appropriate number of wells of the same type of 96-well plate used for planktonic bacteria. The outer wells were filled with 200 μl of sterile PBS to counter dehydration of the biofilms. The plate was cultured aerobically and under static conditions for 24 hours at 37˚C. After incubation the medium was carefully removed by pipetting and the biofilms were washed twice with sterile PBS to remove non-adherent bacteria. Finally 50 μl of sterile PBS was added to each well containing biofilms. All assays were carried out in duplicate.

## Antibodies and radiolabeling

To deliver the radionuclides to the bacteria and biofilms, human mAb anti-β GlcNAc-IgG1 antibody 4497 that targets wall teichoic acids (WTAs) was used [9]. WTAs are cell surface-exposed glycopolymers on *S. aureus* cells that are also found within the extracellular matrix of the biofilm [10]. Thus, this antibody targets both bacteria and biofilm. Human mAb Palivizumab (IgG1) against respiratory syncytial virus (RSV) was acquired from MedImmune and was used as an isotype-matching negative control for non-specific killing of bacteria. Unlabeled 4497 and Palivizumab mAbs were also used as controls. MAbs 4497 and Palivizumab were conjugated to bifunctional chelating agents CHXA" or DOTA (Macrocyclics, USA) using a 10- or 20-fold molar excess over mAb as described earlier [11]. Conjugated antibody was radiolabeled with three different radioisotopes, $^{213}$Bismuth, $^{177}$Lutetium and $^{225}$Actinium. Radiolabeling of the antibody-CHXA" conjugate with $^{177}$Lu and $^{213}$Bi was performed to achieve a specific activity of 185 kBq/μg of the antibody whereas for the radiolabeling of antibody-DOTA conjugate with $^{225}$Ac a specific activity of 37 kBq/μg was desired.

$^{213}$Bi was eluted from the generator with 200 μL of freshly prepared 2% (v/v) HI solution in deionized $H_2O$ followed by 100 μL of deionized $H_2O$. To facilitate radiolabeling, the pH was adjusted to 7 using 80 μL of 5M ammonium acetate solution and the radioactivity was measured on a dose calibrator. An appropriate amount of Ab-CHXA" was then added to achieve the desired specific activity and the reaction was heated at 37˚C for 5 minutes with shaking. The reaction was then quenched by the addition of 3 μL of 0.05 M EDTA solution to bind any free $^{213}$Bi. To purify the mixture the solution was then added to an Amicon Ultra 0.5 mL centrifugal filter (30K MW cut off, Fisher Scientific) and spun for 3 minutes at 14 000 g, followed by 300 uL of PBS was and spun again. The purified solution was collected and the percentage of radiolabeling (radiolabeling yield) was measured by instant thin layer chromatography (iTLC) by developing 10 cm silica gel strips (Agilent Technologies, CA, USA) in 0.15 M ammonium acetate buffer. In this system the radiolabeled antibodies stay at the point of application while free $^{213}$Bi, in the form of EDTA complexes, moves with the solvent front. The strips were cut in half and each half is counted on a 2470 Wizard2 Gamma counter (Perkin Elmer, MA, USA) that was calibrated for the $^{213}$Bi emission spectrum and only emissions in this range were considered in the CPM. The percentage of radiolabeling is calculated by dividing the counts per minute (CPM) at the bottom of the strip (labeled antibody) by the sum of the CPM at the bottom and the top of the strip (total amount of radioactivity) and multiplying the result by 100. Typical yields were greater than 95%.

$^{177}$Lu chloride was diluted with 0.15 M ammonium acetate buffer and added to a microcentrifuge tube (MCT) containing the mAb-CHXA" conjugate in the 0.15 M ammonium acetate buffer in a reaction volume of ~30 μL. The reaction mixture was incubated for 60 min at 37˚C.

The reaction was then quenched by the addition of 3 μL of 0.05 M EDTA solution to bind any free [177]Lu. Labelling efficiency was then measured in the same manner described above using iTLC. Typical yields were greater than 95% and required no further purification.

[225]Ac labelling was performed similarly to [177]Lu, however to accommodate the larger size of [225]Ac an antibody-DOTA conjugate had to be used, Three μL of 0.05M Diethylenetriamine pentaacetate (DTPA) solution was used to quench the reaction. iTLC were read 24h after running to allow for secular equilibrium to be reached. Yields were typically greater than 98% and required no further purification.

### Determination of the bactericidal effect of RIT on *S. aureus*

*S. aureus* AH4802 planktonic cells and biofilms were treated with three different radionuclides at different doses for 1 hour at 37˚C. The doses for [177]Lu were 14.8, 7.4, 3.7 MBq and for [225]Ac 7.4, 3.7 and 1.85 kBq on for both planktonic and biofilm. 370, 185, 111, 74, and 37 kBq of [213]Bi was used on planktonic bacteria and 185, 111 and 37 kBq was used on biofilms. Alpha particles have higher (100 keV/μm) linear energy transfer when compared to beta particles (0.8 keV/μm) and can produce considerably more lethal double strand DNA breaks along their tracks. Therefore, [177]Lu (half-life 6.7 days, 0.5 MeV), being a beta-emitter, needs higher activities to deliver a lethal absorbed dose when compared to alpha-radiation. During its decay, [225]Ac (half-life 10 days, 5.9 MeV) emits four α-particles versus one α-particles emitted by [213]Bi (half-life 46 min, 5.9 MeV). Thus, [225]Ac is lethal at lower activities when compared to [213]Bi and therefore a lower radiation dose of [225]Ac is used.

After incubation, the 96-wells plates were centrifuged for 7 min at 3,500 RPM, washed twice and the pellets were re-suspended in 100 μl sterile PBS. The biofilms were sonicated for 30 seconds to detach the cells. Viability and metabolic activity of the bacterial cells and biofilms were measured by CFU dilution and XTT reduction. For viability testing, 20 μl of each well was used to make a serial dilution, cultured overnight on blood agar plates and counted for colony forming units (CFU). Metabolic activity was measured by adding 50 μl of freshly mixed 2,3-bis-(2-methoxy-4-nitro-5-sulfophenyl)-2H-tetrazolium-5-carboxanilide (XTT) solution (from XTT cell proliferation kit II, Sigma) to the remaining 80 μl of the bacterial solution in each well. The plates are covered in aluminum foil and incubated for 3 hours at 37˚C under static conditions. The colorimetric change was read in an ELISA plate reader (Labsystem Multiskan, Franklin, MA) at 492 nm absorbance. Colorimetric change is the result of mitochondrial dehydrogenase activity that reduced XTT tetrazolium salt to XTT formazan. Thus, the more colorimetric change the more metabolic activity there is. Two wells with sterile PBS were used as blank control. Significant differences between treatment groups and their corresponding controls were calculated using one-way ANOVA with a post-hoc Tukey test.

### Results

Multiple experiments were done to assess the efficacy of RIT on planktonic *MRSA* and biofilms. First, planktonic bacteria were treated with 14,8, 7,4 and 3,7 MBq of beta-emitter [177]Lu bound to the specific WTA-specific mAb 4497 and to the non-specific mAb Palivizumab. XTT results were compared to the unlabeled controls. (Fig 1A and 1B). No significant reduction of cells was seen between treatment groups and their controls (p = >0,05). A significant difference in metabolic activity was seen between the specific and non-specific antibodies loaded with 7,4 MBq of [177]Lu (p = 0.033). Second, planktonic bacteria were treated with 7,4, 3,7, and 1,85 kBq of alpha-emitter [225]Ac bound to the same antibodies. No significant difference was seen in CFU count between the groups apart from the non-specific treatment group bound to 3.7 kBq of [225]Ac compared to the control. (p = >0,05) A significant increase in metabolic

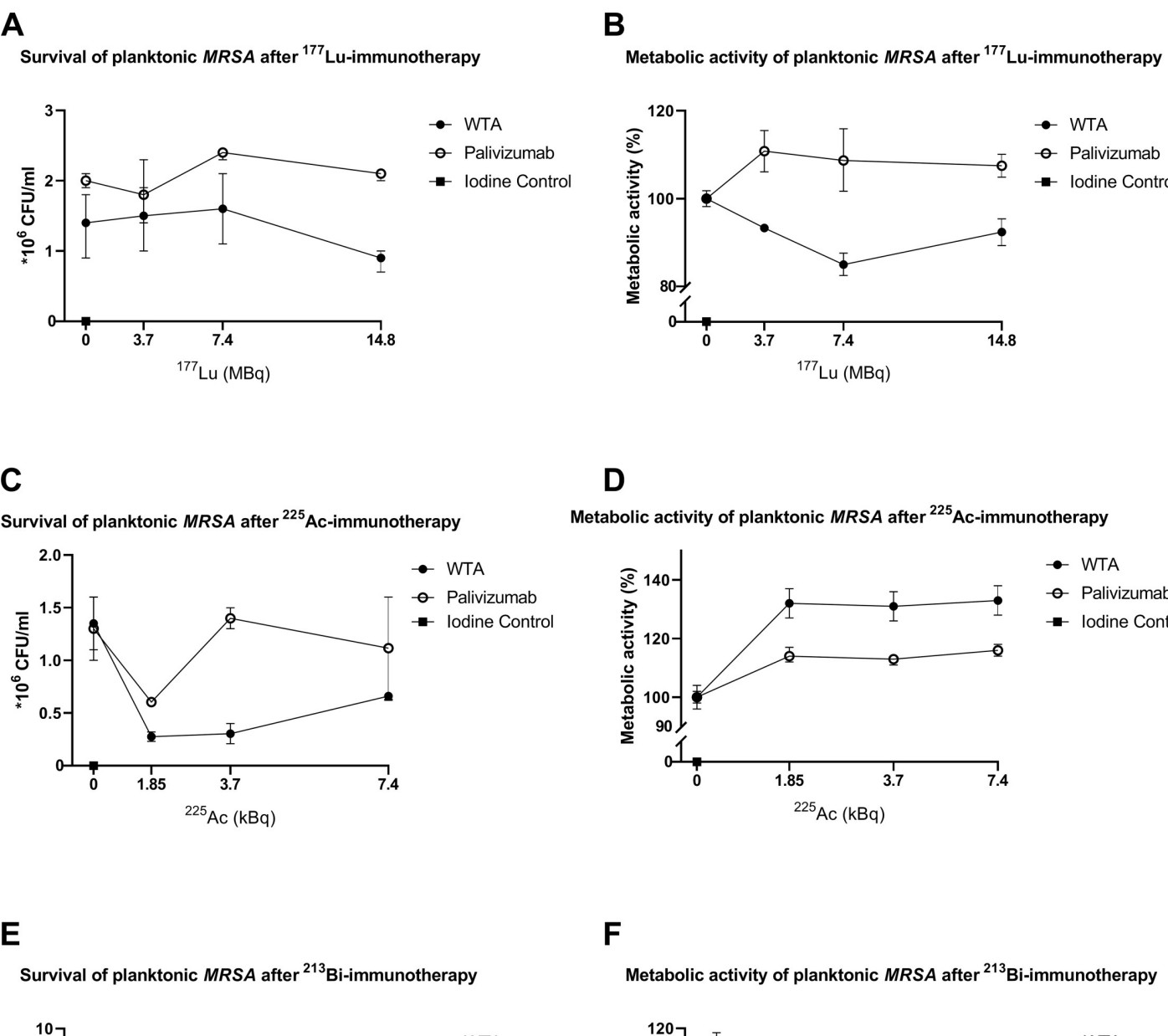

**Fig 1. Susceptibility of <u>planktonic</u> *S. aureus* (*MRSA*) to beta and "short and long-lived" alpha radiation measured by CFU/ml for survivability (A,C,E) and XTT reduction assay for the metabolic activity (B,D,F).** Increasing doses of RIT with specific anti-WTA 4497 antibodies and non-specific antibodies Palivizumab labeled with 177Lu (A,B), 225Ac (C,D) and 213Bi (E,F). Treatment results were compared to unlabeled 4497 mAb,Palivizumab, and iodine controls. Each data point represents the average of two measurements.

activity was seen in all treatment groups compared to both controls and corresponding radiation dose. (p = > 0,05) (Fig 1C and 1D). WTA-specific mAb [225]Ac-4497 showed an increase in metabolic activity when compared the labeled non-specific antibodies. Third, planktonic bacteria were treated with 370, 185, 111, 74 and 37 kBq of [213]Bi. When treatment groups were compared to their controls, a significant reduction in survival was seen in the WTA-specific mAb 4497 group with 370, 185 and 111 kBq (p = 0,002, p = 0,008 resp. p = 0,032) when compared to the control. Also, a significant reduction was seen with the non-specific antibodies 370 and 185 kBq compared to the control (p = 0,003 resp. p = 0,036).

After the planktonic experiments, biofilms were treated with RIT. The WTA-specific mAb 4497 groups with [177]Lu, [225]Ac and [213]Bi showed no significant difference in CFU count compared to the control although there was no bacterial growth with 185 kBq [213]Bi bound to WTA and Palivizumab antibodies (Fig 2). There was a significant difference in metabolic activity between [177]Lu bound to the non-specific antibodies compared to the control (p = 0,026) and a significant difference, in favor of the specific antibody, between the WTA-specific mAb 4497 with a radiation dose of 7.4 MBq when compared to the same dose of the non-specific antibody (p = 0,011). In the non-specific antibody group of [225]Ac with 1.85 kBq, a significant difference was seen in metabolic activity when compared to the control (p = 0,029). A significant difference in metabolic activity was seen in WTA-specific mAb 4497 loaded with 185 and 111 kBq of [213]Bi when compared to the control (p = 0,014 resp. p = 0,045) Also, a significant difference in metabolic activity was seen in the non-specific antibody group loaded with 185 kBq of [213]Bi compared to the control (p = 0,018). There was a significant difference in metabolic activity between the specific and non-specific antibodies loaded with 111 and 74 kBq of [213]Bi (p = 0,037 resp. p = 0,026). There was no significant difference seen in metabolic activity between the specific and non-specific antibodies loaded with the highest dose of 185 kBq [213]Bi because all the plates were sterile.

## Discussion

The need for alternative treatment options for patients with implant infections like periprosthetic joint infections grows every year, not only due to increasing pathogen resistance to antibiotics, but also because biofilm formation obstructs the treatment of these infections with antibiotics. To the best of our knowledge, this is the first report of RIT treatment of MRSA in particular and of any multidrug resistant bacteria in general in both planktonic and biofilm state. [213]Bi-labeled mAb WTA 4497 consistently killed both planktonic bacteria and biofilm as measured by both XTT and CFU assays. A significant decrease in both CFU count and metabolic activity was seen in planktonic bacteria when treated with 370, 185 and 111 kBq 213Bi-WTA-specific mAb 4497 versus 370 and 185 kBq of 213Bi-Palivizumab (Fig 1E and 1F). Although statistically not significant, biofilms treated with specific and non-specific antibodies labeled with 185 kBq of [213]Bi showed no residual bacterial growth with the effect being similar to that of Iodine solution, being used as a positive control. (Figs 1 and 2). Also, a significant decrease in metabolic activity was seen in biofilms treated with specific antibodies labeled with 185 and 111 kBq of [213]Bi and non-specific antibodies labeled with 185 kBq (Fig 2E and 2F). The decrease in survival and metabolic activity in both planktonic bacteria and biofilms treated with 370 and 185 kBq [213]Bi groups was probably due to high levels of radioactivity in a small volume causing non-specific killing. Nonetheless, [213]Bi showed a dose dependent killing of planktonic bacteria and biofilms. Interestingly, specific antibodies labeled with 111 kBq showed significantly more killing of planktonic bacteria and biofilms in both CFU count and XTT when compared to non-specific antibodies loaded with the same amount of radiation, suggesting that specific targeting is more effective. This could mean that mAb [213]Bi-4497

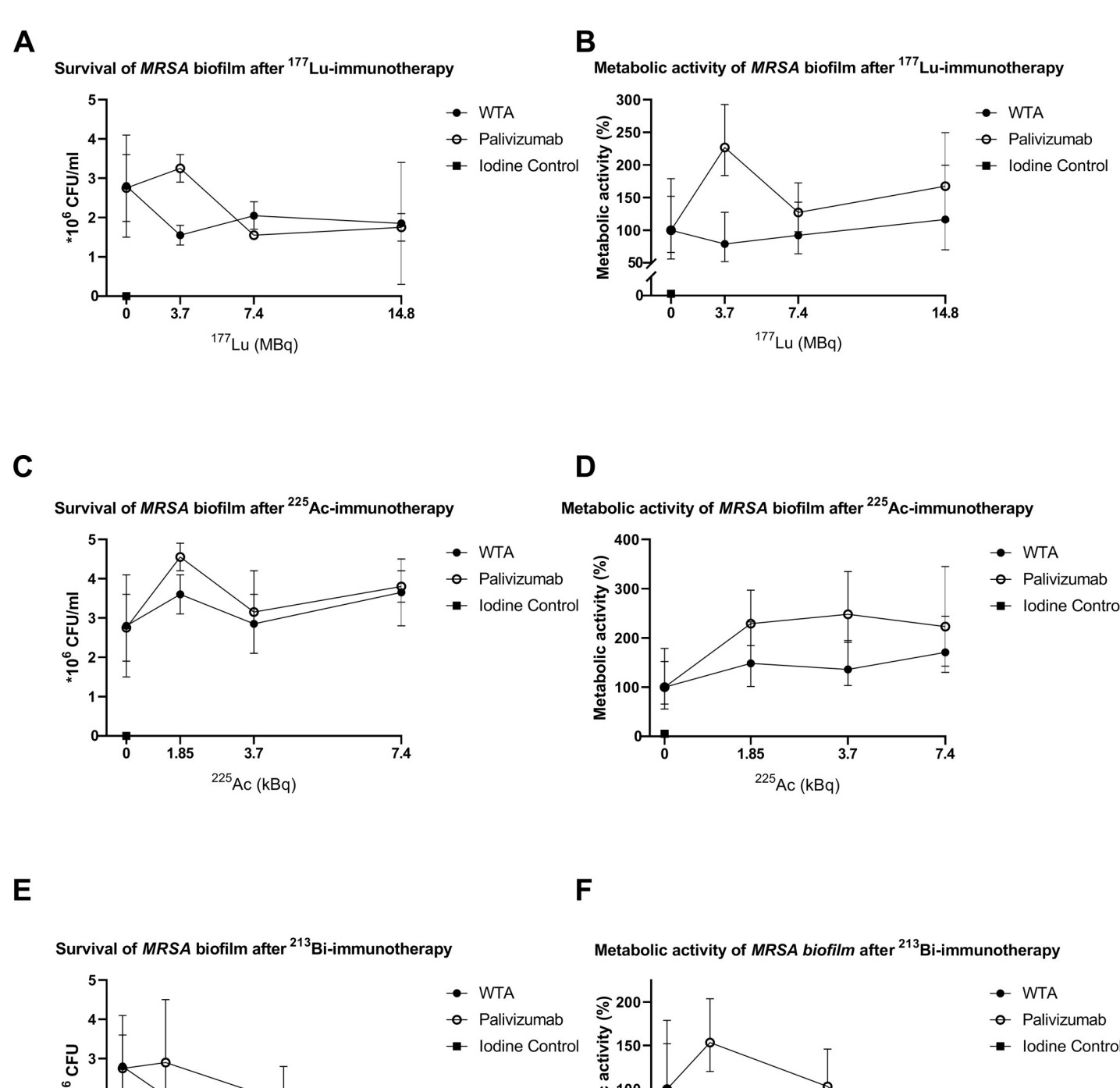

**Fig 2. Susceptibility of *S. aureus* (*MRSA*) biofilm to beta and "short and long-lived" alpha radiation measured by CFU/ml for survivability (A,C,E) and XTT reduction assay for the metabolic activity (B,D,F).** Increasing doses of RIT with specific anti-WTA 4497 antibodies and non-specific antibodies Palivizumab labeled with $^{177}$Lu (A,B), $^{225}$Ac (C,D) and $^{213}$Bi (E,F). Treatment results were compared to unlabeled 4497 mAb, Palivizumab, and iodine controls. Each data point represents the average of two measurements.

specifically targets individual cells. In this regard, the non-specific killing by both radiolabeled bacterium-specific and non-specific antibodies results from one or combination of two events: 1) in vitro there is always some non-specific killing of cells by highly destructive alpha and beta particles emitted by the radionuclides in a small volume of an assay; 2) WTA-specific 4497 and RSV mAbs are human monoclonal antibodies which, on average, have much higher isoelectric points (IP) than murine antibodies. Antibodies with higher IPs have a tendency to non-specifically bind to the cells, therefore, both radiolabeled 4497 and RSV mAbs demonstrated some non-specific therapeutic effect towards bacterial cells. However, the killing effect of radiolabeled with [213]Bi 4497 mAb was higher than that of radiolabeled RSV mAb under examined conditions, indicating that killing of MRSA by [213]Bi-4497 mAb was WTA-specific". This also suggests that α-particles are able to effectively penetrate the architecture of the biofilms to deliver bactericidal radiation to the cells. Encouragingly, the dose required for killing *S. aureus* in a biofilm was of the same order of magnitude as the dose required to kill planktonic cells. [177]Lu-4497 mAb did not have an effect on planktonic bacteria and the biofilm, whereas [225]Ac seemed to even increase metabolic activity in planktonic and biofilm formation. This was probably due to the biofilm matrix release and thus bacterial release, caused by alpha-radiation, which interferes with XTT and gives an increase in CFU when compared to the unlabeled controls. It is possible, that due to the long physical half-lives of [225]Ac (10 days) and [177]Lu (6.7 days) it might take longer than the 1 hour incubation to reveal the full bactericidal potential of these long lived radionuclides but longer incubation time might also improve the damaging effect of [213]Bi.

Previously we have performed extensive evaluation of RIT toxicity in mouse models of fungal and bacterial infections using the antibodies labeled with the same [213]Bi radionuclide utilized in this work. No systemic toxicity was noted in mice infected with *Streptococcus pnemoniae* and treated with [213]Bi-labeled antibody to bacterial polysaccharide [12]. The toxicity evaluation of RIT-treated mice infected intratracheally with *C. neoformans* showed the absence of acute hematologic and long-term pulmonary toxicity [13]. RIT was much better tolerated by the treated mice than Amphoterecin B which is a current standard of care for invasive fungal infections [14]. In addition, RIT did not adversely affect bystander mammalian cells such as CHO cells and macrophages, with the latter being able to carry out their functions such as nitric oxide production after RIT exposure [15]. As all these observations have been made for systemic administration of the radiolabeled antibodies, the anticipated local application of RIT into the infected joint should be even safer in this regard.

In conclusion, our results demonstrate the ability of specific antibodies loaded with an alpha-emitter [213]Bi to selectively kill *S aureus* cells in vitro in both planktonic and biofilm state. RIT could therefore be a potentially alternative treatment modality against planktonic and biofilm-related microbial infections and can be used with and without conventional therapies such as antibiotics. However, this in vitro observed bactericidal effect of RIT on *S. aureus* must be validated in vivo and the work in the animal models of MRSA infection is currently on-going.

## Supporting information

**S1 Data.**
(XLSX)

## Author Contributions

**Conceptualization:** B. van Dijk, K. J. H. Allen, H. C. Vogely, M. G. E. H. Lam, J. M. H. de Klerk, H. Weinans, B. C. H. van der Wal, E. Dadachova.

**Data curation:** B. van Dijk, K. J. H. Allen, M. Helal, B. C. H. van der Wal, E. Dadachova.

**Formal analysis:** B. van Dijk, E. Dadachova.

**Funding acquisition:** H. C. Vogely, H. Weinans, B. C. H. van der Wal, E. Dadachova.

**Investigation:** B. van Dijk, K. J. H. Allen, E. Dadachova.

**Methodology:** B. van Dijk, E. Dadachova.

**Project administration:** B. van Dijk, B. C. H. van der Wal.

**Resources:** K. J. H. Allen.

**Supervision:** H. C. Vogely, H. Weinans, B. C. H. van der Wal, E. Dadachova.

**Validation:** B. van Dijk, B. C. H. van der Wal, E. Dadachova.

**Writing – original draft:** B. van Dijk, H. Weinans, B. C. H. van der Wal, E. Dadachova.

**Writing – review & editing:** B. van Dijk, H. C. Vogely, M. G. E. H. Lam, J. M. H. de Klerk, H. Weinans, B. C. H. van der Wal, E. Dadachova.

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
