## [Editor Report · Decision Letter 0]

8 Jan 2020

PONE-D-19-33702

Radioimmunotherapy of Methicillin-resistant Staphylococcus aureus in planktonic state and biofilms

PLOS ONE

Dear Dr  Dadachova Ekaterina

Thank you for submitting your manuscript to PLOS ONE. After careful consideration, we feel that it has merit but does not fully meet PLOS ONE’s publication criteria as it currently stands. Therefore, we invite you to submit a revised version of the manuscript that addresses the points raised during the review process.

Though the manuscript is presented as a proof of concept, we recommend that the potential toxicity of such treatment to be highlighted in the manuscript. Accordingly, you are advised to modify the manuscript highlighting this limitation and resubmit . 

We would appreciate receiving your revised manuscript by Feb 22 2020 11:59PM. To enhance the reproducibility of your results, we recommend that if applicable you deposit your laboratory protocols in protocols.io, where a protocol can be assigned its own identifier (DOI) such that it can be cited independently in the future. For instructions see: http://journals.plos.org/plosone/s/submission-guidelines#loc-laboratory-protocols

We look forward to receiving your revised manuscript.

Kind regards,

Amal Al-Bakri

Academic Editor

PLOS ONE

Additional Editor Comments:

Dear Authors,

Though the manuscript is presented as a proof of concept, we recommend that the potential toxicity of such treatment to be highlighted in the manuscript. Accordingly, you are advised to modify the manuscript highlighting this limitation and resubmit .

Regards

Journal Requirements:

2. In your Methods section, please give the sources of any bacterial strains used in your study.

6. Thank you for stating the following in the Financial Disclosure  section:

"This research was funded by Health Holland"

We note that you received funding from a commercial source: Health Holland

d. Please include both your amended Competing Interests and Funding Statements within your cover letter. We will change the online submission form on your behalf.
---

## [Author Response · Author response to Decision Letter 0]

3 Feb 2020

Reviewer 1

Though the manuscript is presented as a proof of concept, we recommend that the potential toxicity of such treatment to be highlighted in the manuscript. Accordingly, you are advised to modify the manuscript highlighting this limitation and resubmit. 

Response: We have added the following paragraph and additional references 12-15 to the revised manuscript. The paragraph reads: “Previously we have performed extensive evaluation of RIT toxicity in mouse models of fungal and bacterial infections using the antibodies labeled with the same 213Bi radionuclide utilized in this work. No systemic toxicity was noted in mice infected with Streptococcus pnemoniae and treated with 213Bi-labeled antibody to bacterial polysaccharide (12). The toxicity evaluation of RIT-treated mice infected intratracheally with C. neoformans showed the absence of acute hematologic and long-term pulmonary toxicity (13). RIT was much better tolerated by the treated mice than Amphoterecin B which is a current standard of care for invasive fungal infections (14). In addition, RIT did not adversely affect bystander mammalian cells such as CHO cells and macrophages, with the latter being able to carry out their functions such as nitric oxide production after RIT exposure (15). As all these observations have been made for systemic administration of the radiolabeled antibodies, the anticipated local application of RIT into the infected joint should be even safer in this regard.”

Journal Requirements

Response: We have modified the manuscript according to PLOS ONE style. 

2. In your Methods section, please give the sources of any bacterial strains used in your study.

Response: We have included the source of bacteria into the Methods 

3. We note that you have indicated that data from this study are available upon request. PLOS only allows data to be available upon request if there are legal or ethical restrictions on sharing data publicly. We will update your Data Availability statement on your behalf to reflect the information you provide. – Response: We have uploaded the data as a Supporting information file.

4. PLOS requires an ORCID iD for the corresponding author in Editorial Manager on papers submitted after December 6th, 2016. Please ensure that you have an ORCID iD and that it is validated in Editorial Manager. To do this, go to ‘Update my Information’ (in the upper left-hand corner of the main menu), and click on the Fetch/Validate link next to the ORCID field. This will take you to the ORCID site and allow you to create a new iD or authenticate a pre-existing iD in Editorial Manager. Please see the following video for instructions on linking an ORCID iD to your Editorial Manager account: https://www.youtube.com/watch?v=_xcclfuvtxQ – Response:

ORCID iD has been linked to the manuscript.

5. We note that you have included the phrase “data not shown” in your manuscript. Unfortunately, this does not meet our data sharing requirements. PLOS does not permit references to inaccessible data. We require that authors provide all relevant data within the paper, Supporting Information files, or in an acceptable, public repository. Please add a citation to support this phrase or upload the data that corresponds with these findings to a stable repository (such as Figshare or Dryad) and provide and URLs, DOIs, or accession numbers that may be used to access these data. Or, if the data are not a core part of the research being presented in your study, we ask that you remove the phrase that refers to these data. – Response: We are now showing this data in the modified figures.

---

## [Decision Letter · Decision Letter 1]

15 Apr 2020

PONE-D-19-33702R1

Radioimmunotherapy of Methicillin-resistant Staphylococcus aureus in planktonic state and biofilms

PLOS ONE

Dear Dr Dadachova

Thank you for submitting your manuscript to PLOS ONE. After careful consideration, we feel that it has merit but does not fully meet PLOS ONE’s publication criteria as it currently stands. Therefore, we invite you to submit a revised version of the manuscript that addresses the points raised during the review process.

Specifically, reviewer recommends more experimental work to be done.

We would appreciate receiving your revised manuscript by May 30 2020 11:59PM. To enhance the reproducibility of your results, we recommend that if applicable you deposit your laboratory protocols in protocols.io, where a protocol can be assigned its own identifier (DOI) such that it can be cited independently in the future. For instructions see: http://journals.plos.org/plosone/s/submission-guidelines#loc-laboratory-protocols

We look forward to receiving your revised manuscript.

Kind regards,

Amal Al-Bakri

Academic Editor

PLOS ONE

Reviewers' comments:

Reviewer's Responses to Questions

**Comments to the Author**

1. If the authors have adequately addressed your comments raised in a previous round of review and you feel that this manuscript is now acceptable for publication, you may indicate that here to bypass the “Comments to the Author” section, enter your conflict of interest statement in the “Confidential to Editor” section, and submit your "Accept" recommendation.

Reviewer #1: All comments have been addressed

Reviewer #2: All comments have been addressed

2. Is the manuscript technically sound, and do the data support the conclusions?

Reviewer #1: Yes

Reviewer #2: Yes

3. Has the statistical analysis been performed appropriately and rigorously? 

Reviewer #1: Yes

Reviewer #2: Yes

4. Have the authors made all data underlying the findings in their manuscript fully available?

Reviewer #1: Yes

Reviewer #2: Yes

5. Is the manuscript presented in an intelligible fashion and written in standard English?

Reviewer #1: Yes

Reviewer #2: Yes

6. Review Comments to the Author

Reviewer #1: This manuscript expands on the growing repertoire of immunotherapy applications, outside of cancer. The authors have provided a proof-of-concept study investigating radioimmunotherapy for the treatment of Methicillin-resistant staphylococcus aureus. As such, rigorous experimentation as been performed and the data provided is sufficient. This research indeed contributes to a gap in the field of therapeutic options for prosthetic joint infection (PJI) and offers a novel approach. No additional revision is required, as authors have adequately addressed the comments of previous reviewer. Furthermore, this manuscript adheres to PLOS ONE criteria.

Reviewer #2: The authors have addressed the issues raised in the previous review. However, there is little innovation here. It is not clear that there is a specific antibody effect and whatever was observed would be expected. An in vivo study would generate more interest.

7. PLOS authors have the option to publish the peer review history of their article (what does this mean?). If published, this will include your full peer review and any attached files.

Reviewer #1: Yes: Dr. Krupa Naran

Reviewer #2: No

---

## [Author Response · Author response to Decision Letter 1]

20 Apr 2020

Reviewer #2: 

The authors have addressed the issues raised in the previous review. However, there is little innovation here. It is not clear that there is a specific antibody effect and whatever was observed would be expected. An in vivo study would generate more interest. – Response: We would like to thank the Reviewer for acknowledging that we have addressed the issues in the previous review. 

We respectfully disagree about the lack of the innovation. In the revised manuscript we have expanded the sentence in the Discussion stating that: “To the best of our knowledge, this is the first report of RIT treatment of MRSA in particular and of any multidrug resistant bacteria in general in both planktonic and biofilm state”.

We have also added the explanation about some non-specfic killing to the Discussion: “In this regard, the non-specific killing by both radiolabeled bacterium-specific and non-specific

antibodies results from one or combination of two events: 1) in vitro there is always some non-specific killing of cells by highly destructive alpha and beta particles emitted by the radionuclides in a small volume of an assay; 2) WTA-specific 4497 and RSV mAbs are human monoclonal antibodies which, on average, have much higher isoelectric points (IP) than murine antibodies. Antibodies with higher IPs have a tendency to non-specifically bind to the cells, therefore, both radiolabeled 4497 and RSV mAbs demonstrated some non-specific therapeutic effect towards bacterial cells. However, the killing effect of radiolabeled with 213Bi 4497 mAb was higher than that of radiolabeled RSV mAb under examined conditions, indicating that killing of MRSA by 213Bi-4497 mAb was WTA-specific”.

In regard to the animal experiments, they are out of scope of this proof of principle manuscript.

Currently on-going work in animal models on non-invasive nuclear imaging, therapy and safety evaluation will be reported in a follow-up manuscript. We have added a sentence to the conclusions that “the work in the animal models of MRSA infection is currently on-going”.

---

## [Decision Letter · Decision Letter 2]

29 Apr 2020

Radioimmunotherapy of Methicillin-resistant Staphylococcus aureus in planktonic state and biofilms

PONE-D-19-33702R2

Dear Dr. Dadachova, 

We are pleased to inform you that your manuscript has been judged scientifically suitable for publication and will be formally accepted for publication once it complies with all outstanding technical requirements.

With kind regards,

Amal Al-Bakri

Academic Editor

PLOS ONE

Additional Editor Comments (optional):

Reviewers' comments:

Reviewer's Responses to Questions

**Comments to the Author**

1. If the authors have adequately addressed your comments raised in a previous round of review and you feel that this manuscript is now acceptable for publication, you may indicate that here to bypass the “Comments to the Author” section, enter your conflict of interest statement in the “Confidential to Editor” section, and submit your "Accept" recommendation.

Reviewer #2: All comments have been addressed

2. Is the manuscript technically sound, and do the data support the conclusions?

Reviewer #2: Yes

3. Has the statistical analysis been performed appropriately and rigorously? 

Reviewer #2: Yes

4. Have the authors made all data underlying the findings in their manuscript fully available?

Reviewer #2: Yes

5. Is the manuscript presented in an intelligible fashion and written in standard English?

Reviewer #2: Yes

6. Review Comments to the Author

Reviewer #2: While the reviewers did not provide in vivid data, I think the concept is novel enough to warrant publication in the hopes that in vivid data will be forthcoming

7. PLOS authors have the option to publish the peer review history of their article (what does this mean?). If published, this will include your full peer review and any attached files.

Reviewer #2: Yes: Malik E Juweid

---

## [Editor Report · Acceptance letter]

4 May 2020

PONE-D-19-33702R2 

Radioimmunotherapy of Methicillin-resistant *Staphylococcus aureus* in planktonic state and biofilms 

Dear Dr. Dadachova:

I am pleased to inform you that your manuscript has been deemed suitable for publication in PLOS ONE. Congratulations! Your manuscript is now with our production department. 

With kind regards,

on behalf of

Dr. Amal Al-Bakri 

Academic Editor

PLOS ONE